# Nonlinear visuoauditory integration in the mouse superior colliculus

**Shinya Ito**[1,2], **Yufei Si**[3], **Alan M. Litke**[1], **David A. Feldheim**[3]*

**1** Santa Cruz Institute for Particle Physics, University of California, Santa Cruz, California, United States of America, **2** Mindscope program, Allen Institute, Seattle, Washington, United States of America, **3** The Department of Molecular, Cell and Developmental Biology, University of California, Santa Cruz, California, United States of America

* dfeldhei@ucsc.edu

## Abstract

Sensory information from different modalities is processed in parallel, and then integrated in associative brain areas to improve object identification and the interpretation of sensory experiences. The Superior Colliculus (SC) is a midbrain structure that plays a critical role in integrating visual, auditory, and somatosensory input to assess saliency and promote action. Although the response properties of the individual SC neurons to visuoauditory stimuli have been characterized, little is known about the spatial and temporal dynamics of the integration at the population level. Here we recorded the response properties of SC neurons to spatially restricted visual and auditory stimuli using large-scale electrophysiology. We then created a general, population-level model that explains the spatial, temporal, and intensity requirements of stimuli needed for sensory integration. We found that the mouse SC contains topographically organized visual and auditory neurons that exhibit nonlinear multisensory integration. We show that nonlinear integration depends on properties of auditory but not visual stimuli. We also find that a heuristically derived nonlinear modulation function reveals conditions required for sensory integration that are consistent with previously proposed models of sensory integration such as spatial matching and the principle of inverse effectiveness.

## Author summary

Integrating visual and auditory stimuli that occur at the same location and time improves our ability to identify and respond to external events. This type of sensory integration is an important brain function, and its deficits are known symptoms of patients with autism and schizophrenia. The Superior Colliculus (SC) is a midbrain structure that plays a critical role in integrating visual and auditory inputs to assess saliency and promote action. Although the response properties of the individual SC neurons to visuoauditory stimuli have been characterized, little is known about the spatial and temporal dynamics of their integration at the population level. Here we recorded the response properties of SC neurons to spatially restricted visual and auditory stimuli using large-scale electrophysiology. We found that the mouse SC contains topographically organized visual and auditory neurons that exhibit nonlinear multisensory integration. We also find that a heuristically derived nonlinear modulation function reveals conditions required for sensory integration that are consistent with previously proposed models of sensory integration. These

**Data Availability Statement:** The data and the code used for this article are available on Figshare (https://doi.org/10.6084/m9.figshare.16750508).

**Funding:** This work was supported by Brain Research Seed Funding provided by UCSC, the

National Institutes of Health (2R01EY022117, 1R21EY032230-01, DAF), and a donation from John Chen to AML. The funders had no role in study design, data collection and analysis, decision to publish, or preparation of the manuscript.

**Competing interests:** The authors have declared that no competing interests exist.

results open the door to further studies using mice that are designed to determine the circuitry underlying multisensory integration, and the mechanisms used for its development.

## Introduction

Integrating stimuli from different modalities that occur at the same location and time improves our ability to identify and respond to external events[1]. The superior colliculus (SC, homologous to the tectum in lower vertebrates) contains a topographic representation of visual space in its superficial layers (sSC) that is aligned with maps of auditory space in its deeper layers (dSC), and has been used as a model to elucidate general principles of how input from visual and auditory sources integrate [1–3]. For example, the barn owl optic tectum has been a model to elucidate the mechanisms used during development to align the visual and auditory spatial maps[4,5] as well as to determine the circuitry used to modulate visuoauditory responses[6,7]. Studies primarily using anesthetized cats have shown that the mammalian dSC also contains neurons that respond to both visual and auditory stimuli (visuoauditory neurons)[8]. Some of these neurons exhibit superlinear multisensory enhancement, a type of multisensory integration (MSI) whereby a neuron's response to a combination of stimuli is greater than the linear sum of the individual unimodal responses[8–14]. These studies, performed at the single neuron level, have determined three features of the stimuli that are required to induce superlinear multisensory enhancement: 1) 'Spatial matching', whereby both stimuli need to come from the same direction in the sensory field; 2) 'temporal matching', whereby the visual and auditory stimuli must overlap in time; and 3) 'inverse effectiveness', in which multisensory enhancement is weakened when the strength of the stimuli is increased[1,8].

While studies based on the owl tectum and cat SC have many advantages, these models are currently limited in their ability to manipulate genes and the activity of neurons, both of which can be used to determine the underlying circuitry that promotes sensory integration and elucidate the mechanisms by which it develops. The mouse is well suited for such analysis, but it is not known if the mouse SC contains visuoauditory neurons that exhibit MSI. Here we used high-density silicon probe recordings in awake mice, presented with simultaneous visual and auditory stimuli, to characterize the MSI properties of mouse SC neurons. We found that the receptive fields (RFs) of visual/auditory bimodal neurons in the dSC overlap and are organized into a topographic map of azimuthal space with some bimodal neurons exhibiting MSI. We also found that the auditory responses of these neurons have early and late temporal components, with only the late component being non-monotonic to the stimulus intensity. To determine the stimuli needed to invoke MSI and the properties of integration, we created a population-level model of the visuoauditory responses of simultaneously recorded individual neurons in which nonlinearity is given to neurons through a single modulation function that is shared by all the neurons in a dataset. The resulting modulation function quantifies the conditions needed for MSI and recapitulates the spatial matching requirements of the stimuli needed to invoke MSI, as well as the inverse effectiveness principle.

## Results

### The receptive fields of visual/auditory neurons in the deep SC are aligned and form a topographic map of space

In order to determine how visual and auditory information is organized in the mouse SC we used silicon probe electrophysiology to record the spiking activity of SC neurons in awake

behaving CBA/CaJ mice in response to visual and auditory spot stimuli, and then determined their spatial RFs to each stimulus (Fig 1A and 1B). We recorded a total of 1281 neurons from 8 mice. The spatial distribution and the fraction of cells that are responsive to visual stimuli, auditory stimuli, and both are shown in Fig 1C and 1D. Consistent with what is known about the location of visual and auditory neurons in the mouse SC [15–19], we find that visually responsive neurons are located in both the superficial and deep SC while auditory responsive neurons are found only in the deep ($\gtrsim$ 400 μm) SC. Many bimodal neurons have overlapping spatial RFs. Fig 1E–1G shows spatially matching visual and auditory RFs of an example neuron. When we plot the RF azimuth vs. SC anteroposterior (AP) location of the neurons, we find that both visually and auditory responsive neurons exhibit a strong correlation between their physical location and their RF location; the visual map (Fig 2A) has a (77 ± 3) °/mm slope and (-1.6 ± 1.9) ° offset, while the auditory map (Fig 2B) has a (73 ± 8) °/mm slope and (10 ± 5) ° offset. This demonstrates that mice, like other mammals, have aligned visual and auditory maps of azimuthal space in the SC [4,18,19].

When we compared the visual and auditory RFs of the 62 bimodal neurons identified in our recordings, we found that (95 ± 3) % (59) of them have overlapping visual and auditory RFs along the azimuth if the RF boundary is defined as 1 σ (Fig 2D). For (76 ± 5) % (47) of these neurons, the auditory RF is larger than the visual RF. The correlation coefficient between the visual RF centers and auditory RF centers of all visuoauditory neurons is 0.77 (Fig 2C). The bimodal neurons only appear in the dSC where most neurons have relatively large visual RFs ($\gtrsim$ 10 °) and their visual/auditory RF azimuthal extent does not correlate with the depth of the cells (Fig 2E and 2F). We did not find properties of bimodal neurons that depend on their location along the depth of the SC. These results demonstrate that the mouse dSC contains neurons that receive both visual and auditory information from the same azimuthal direction, and that they are organized as a topographic map along the A-P axis of the dSC.

## Unimodal visual and early auditory SC responses are monotonic; late auditory SC responses can be non-monotonic

As a first step to understand how visual and auditory responses interact, we stimulated the mouse with different contrast or sound levels and analyzed the temporal structures and intensity tuning of unimodal visual and auditory responses (Fig 3A and 3B). Similar to what has been reported in the ferret [20], we found that auditory SC responses can be temporally segregated into early ($<$ 20 ms) and late ($>$ 20 ms) components (Fig 3B and 3C; example rasters are in S2 Fig), while visual responses occur only in the late time segment (Fig 3A and Table 1). When we measured the intensity tuning curves of the early and late responses, we found that all of the visual (100% (178 / 178)) and nearly all ((99.7 ± 0.3) % (357 / 358)) of the early auditory responses are monotonic in nature, meaning the firing rate of the neuron increases with increasing stimulus intensity (Fig 3D, 3G, 3E and 3H). However, the late auditory responses of some neurons ((11.4 ± 1.9) % (33 / 290)) were non-monotonic (Fig 3F and 3I). Here, we call a response 'non-monotonic' when any of the non-maximum strength stimuli elicit a significantly ($p < 0.001$, ANOVA) higher firing rate than the maximum strength stimulus (non-monotonic cells are marked as red in Fig 3G–3I), and otherwise 'monotonic' (these may not satisfy a strict mathematical condition of monotonicity; however, our focus here is to highlight clear non-monotonic responses of a small set of neurons). The temporal profiles of visual and auditory responses demonstrate that auditory signals arrive at the SC earlier than visual signals; therefore, if the visual and auditory inputs are detected by the retina and ears at the same time, the SC circuit may already be conditioned (modulated) by the early auditory processing

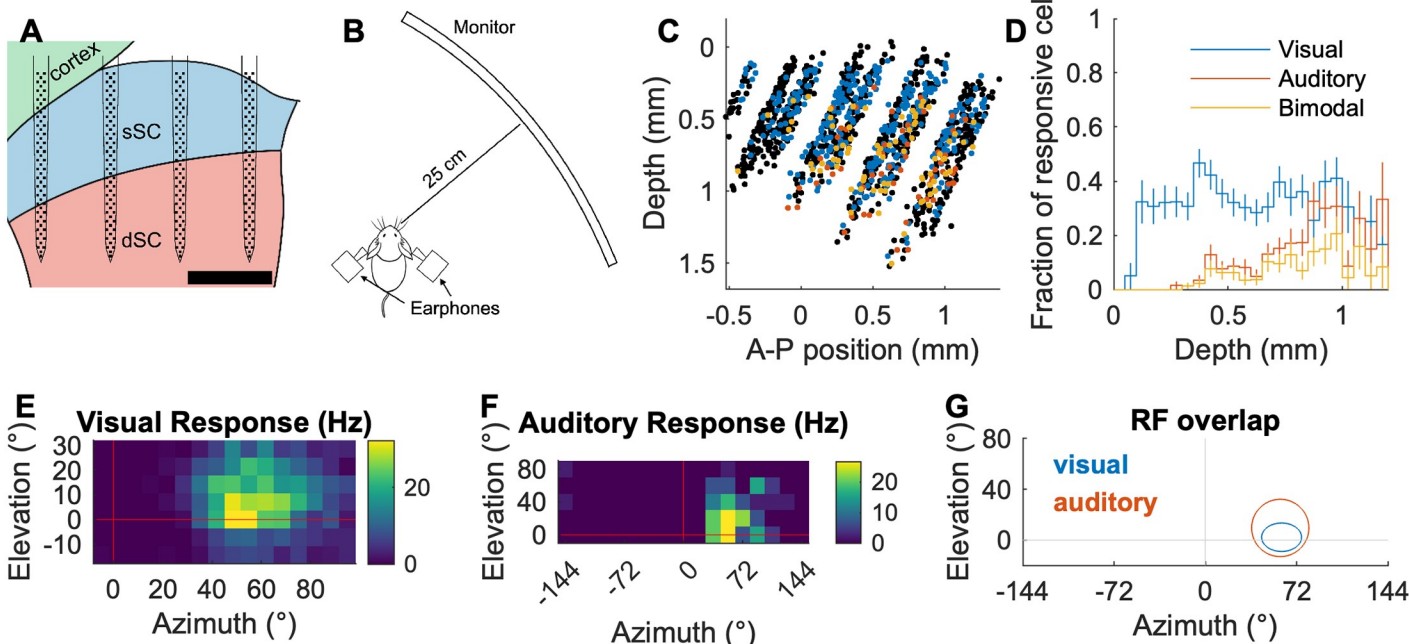

**Fig 1. Experimental setup and RF characterization.** A: A schematic of a 256-electrode silicon probe in the superior colliculus (SC; sagittal plane). sSC: superficial SC, dSC: deep SC. The scale bar is 500 μm. B: A schematic of the recording setup. An awake head-fixed mouse is placed on a cylindrical treadmill. A computer display monitor is placed 25 cm away from the mouse. Virtual auditory space stimuli are delivered through earphones. C: Reconstructed 2D map of all the cells with visual and/or auditory responses (blue: visual, red: auditory, yellow: bimodal, black: non-responsive; significant response with p < 0.001; quasi-Poisson statistics[17]). D: Fraction of the visual/auditory/bimodal neurons as a function of SC depth. E-G: Spatial heat map of response firing rates of an example neuron that shows visual (E) and auditory (F) RFs, and a summary of their overlap (G). Ellipses in (G) indicate 1σ contours of 2D Gaussian fits.

(either in the SC or elsewhere) before the visual input arrives (S1 Fig). This modulation could also explain the non-monotonicity of the late auditory responses.

## Modeling the multisensory responses reveals an audition-dependent modulation structure

The above results are consistent with the hypothesis that auditory signals processed in the first 20ms influence nonlinear processing that occurs later (>20ms). If this hypothesis is correct, nonlinear multisensory processing should depend on the properties of the auditory, but not the visual stimuli. Testing this hypothesis requires detailed analysis of the response in high dimensional stimulus space. Simple approaches such as single post-stimulus time histogram analysis, or significance tests that rely on responses to single patterns are not effective. Therefore, to determine how the early and late responses interact, we used a maximum-likelihood fit analysis of models that include a simple sigmoidal response plus an auditory or visual stimulation-dependent modulation term and asked which model better fits the data.

Briefly, the model illustrated in Fig 4 (see Materials and methods) consists of a baseline firing rate (b) plus sigmoidal visual and auditory responses (S), multiplied by a nonlinearity term (modulation coefficient: $\alpha^\delta$) that depends on the properties of the auditory stimulus. With this formulation, the nonlinear multisensory interaction is summarized in $\alpha^\delta$ and its effect on the response (R) can be parameterized from multiple stimulus patterns. The global modulation function $\delta$ is defined for each mouse, and applied to all the neurons in the mouse, and in this way, it captures the spatial extent of the modulation in the SC circuitry (see Materials and methods for details).

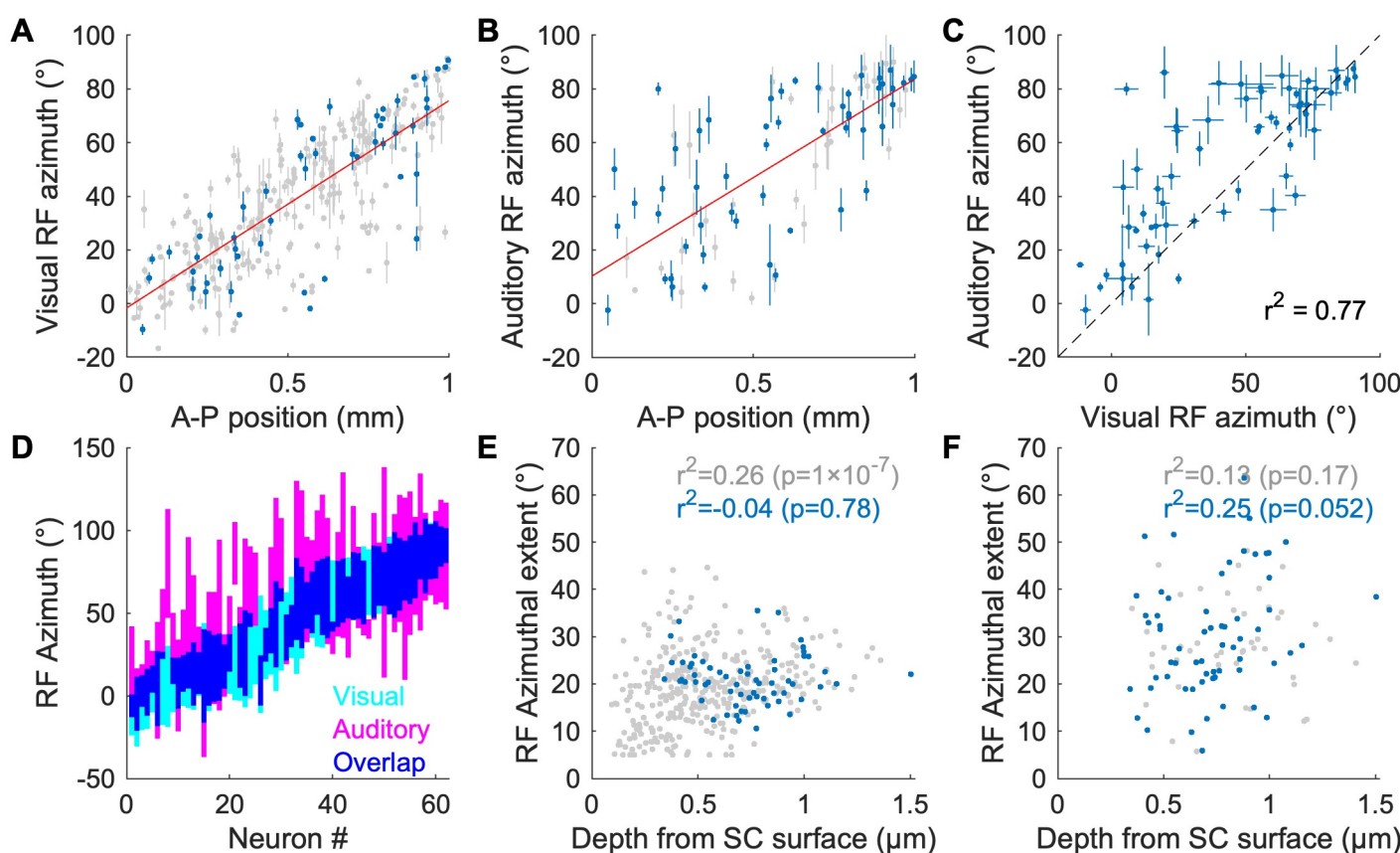

**Fig 2. Topographic organization and receptive field properties of visual and auditory responsive neurons in the SC.** A, B: Scatter plots of estimated RF azimuth vs. A-P SC position for visually responsive (A) and auditory responsive (B) SC neurons. Blue points are neurons that respond to both visual and auditory stimuli; gray points are neurons that respond only to either visual (A) or auditory (B) stimuli. The red lines are linear fits. The error bars represent 1-σ errors (68% CIs) of the estimated azimuths inferred from function fitting. C: A scatter plot of visual and auditory RF azimuths for bimodal neurons. D: RF azimuthal extent of multimodal neurons. The neurons are sorted by their visual RF azimuth. The colored regions are ±1-σ width of the fitted Gaussians (visual RF) or corresponding radius of the Kent distribution (auditory RF; [17]). E, F: Scatter plots of the depth of the neurons and RF azimuthal extent for visually responsive (E) and auditory responsive (F) neurons. Blue points are neurons that respond to both visual and auditory stimuli; gray points are neurons that respond only to either visual (E) or auditory (F) stimulus.

Fig 5 illustrates the example data and the fitting of the model. Fig 5A–5E summarizes the responses of a representative neuron to different levels of visual and auditory stimuli coming from a specific azimuth. The firing rates in the late integration window (Fig 5C) is compared with the linear additions of the unisensory stimuli (Fig 5D). For this neuron, the multisensory firing rates are higher than the addition of the unisensory responses when the auditory stimulus is weak (intensities 1 and 2, that correspond to 20 and 30 dB; exhibiting nonlinear multisensory enhancement (MSE)), and lower when the auditory stimulus is strong (intensities 3 and 4, that correspond to 40 and 50 dB; exhibiting nonlinear multisensory suppression (MSS)). The model recapitulates both MSE and MSS, including its dependency on the auditory intensity (Fig 5G–5I).

To test the hypothesis that nonlinear multisensory processing depends on the properties of the auditory stimuli, we modeled the nonlinearity term as a function of the auditory stimulus properties (audition dependent modulation; AM-model), the visual stimulus properties (vision dependent modulation; VM-model), or fixed to 1 (Linearly added monotonic sigmoids; L-model), and tested how well each model describes the data. To quantify the goodness-of-fit of

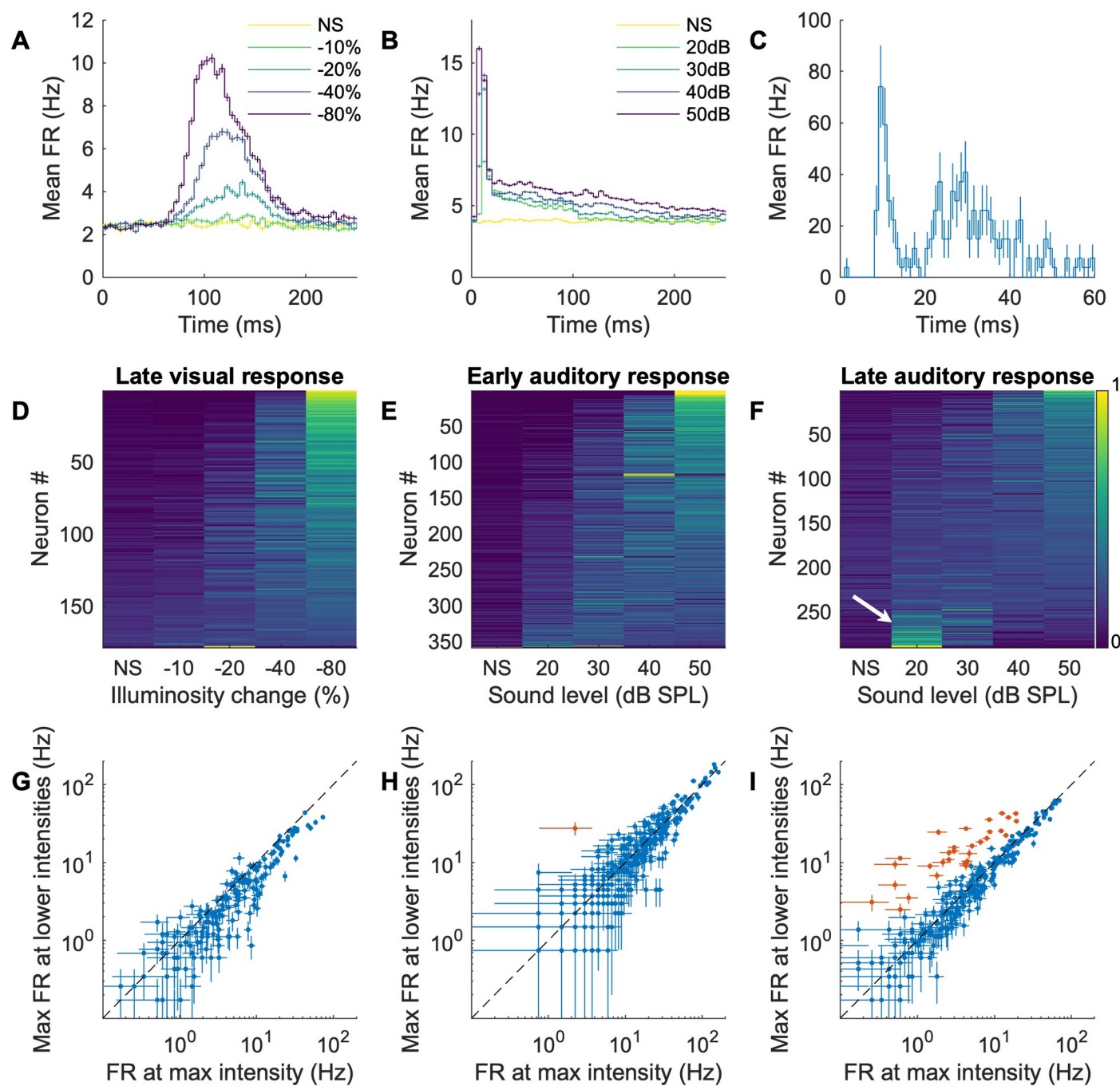

**Fig 3. Temporal segregation and intensity profiles of SC visual and auditory responses.** A, B: Average responses over time of all (A) visually or (B) auditory responsive neurons. Darker colors indicate higher contrast (visual stimuli) or stronger intensities (auditory stimuli; NS: no stimulus). C: The auditory response of an example neuron over time (averaged over 30 trials at 85 grid locations) that has distinct early (<20ms) and late (>20ms) components. The error bars are SEM over the trials. D: Normalized responses to different visual intensities. The neurons are sorted by the center of mass of the intensity response (D-F). E: Normalized early responses to different auditory intensities. F: Normalized late responses to different auditory intensities. Note that a fraction of cells prefer low intensities over high intensities (white arrow). G-I: Scatter plots of the maximum (max) FR at non-max intensity vs the FR at max intensity (red points are non-monotonic neurons: y > x with p < 0.001) of (G) the visual stimulus response; (H) the early auditory response; and (I) the late response to auditory stimuli. The error bars are SEM over 30 trials.

**Table 1. Percentage of neurons that respond to visual and auditory unimodal stimuli.**

| Response type | Number of neurons | Percentage (%) |
| --- | --- | --- |
| Visual early | 1 | 0.08 ± 0.08 |
| Visual late | 178 | 13.9 ± 1.0 |
| Auditory early | 358 | 27.9 ± 1.3 |
| Auditory late | 290 | 22.6 ± 1.2 |
| Auditory early & late | 170 | 13.3 ± 0.9 |
| Visual & auditory late | 65 | 5.1 ± 0.6 |
| Visual or auditory late (Used for modeling) | 403 | 31.5 ± 1.3 |

these models, we used the deviance per degree of freedom (D/DF; this value becomes asymptotically close to the reduced $\chi^2$ in the limit of a large data sample (Wilk's theorem); a D/DF = 1 is a perfect fit; the outlier deviance, D/DF—1, provides quantification of the data that the model does not describe, see Materials and methods). We calculated the D/DF of each model and summarized it in Fig 6A. If we think of the L-model as the baseline, the AM-model exhibited a dramatic improvement of the outlier deviance (0.72 to 0.173), while the VM-model does not (0.72 to 0.66). This confirms the above hypothesis that the nonlinear processing of multisensory responses is better described by the modulatory properties of auditory compared to visual stimuli.

The modulation function of the model provides a quantitative description of the multisensory integration response properties for all neurons in the data set under a wide variety of auditory stimuli. Fig 6B and 6C shows the average shape of the modulation function from 8 datasets (those for each dataset are shown in S3 Fig) and has two important features. First, it has a bounded spatial distribution with a positive center and a negative surround. Second, more intense auditory stimuli cause stronger inhibition than less intense auditory stimuli. These two properties result in the modulation function being positive only near the location of the auditory stimulation, and only if the auditory stimulus is weak. The first property is known

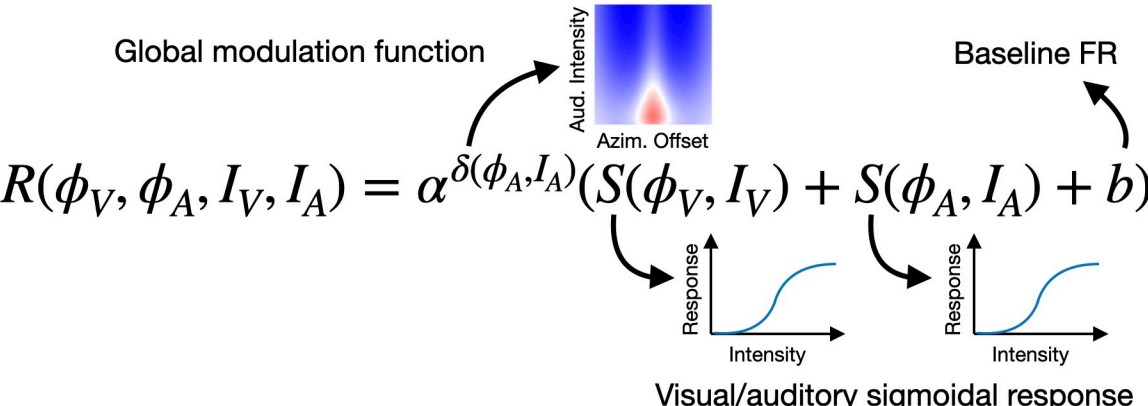

**Fig 4. The model function of visuo-auditory multisensory integration.** This is the equation of the model for audition-dependent modulation. R is response firing rate; $\phi_V$, $\phi_A$ are azimuths of visual and auditory stimuli, respectively; $I_V$, $I_A$ are intensity of the visual or auditory stimuli, respectively; $\alpha$ is a modulation weight for a neuron; $\delta$ is a global modulation function for a mouse; S is a sigmoidal function; and b is a baseline firing rate. See Materials and methods for details. For the vision-dependent model, the global modulation function $\delta$ depends on $\phi_V$ and $I_V$. For the no-modulation model, $\alpha$ is fixed to 1.

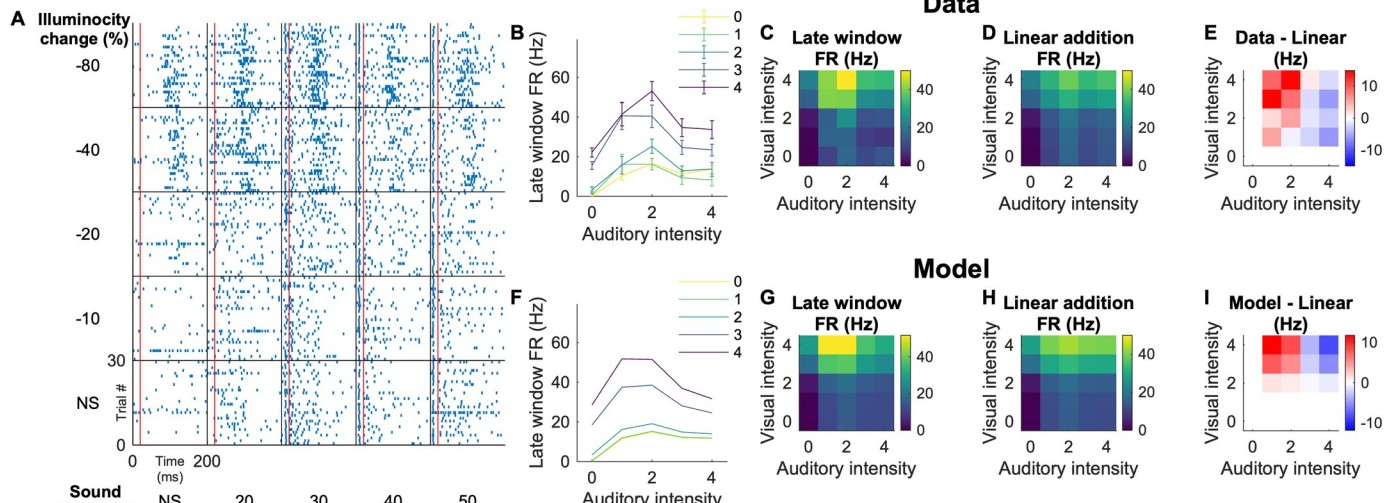

**Fig 5. An example neuron that exhibits both superlinear and sublinear multisensory responses.** A: Raster plots for different combinations of visual contrasts and auditory intensities (NS represents no stimulus). The red lines indicate 20 ms that separates the early and late time window for the auditory response. The auditory RF azimuthal extent for this neuron was 0˚ to 54˚, and the visual RF azimuthal extent was -6˚ to 24˚. These are responses to stimuli that were given at 20˚ in azimuth (both visual and auditory). B: Measured firing rates in the late time window. The colors indicate visual contrasts. The error bars are SEM over 30 trials. The stimulus intensities are indicated as 0–4 for brevity; these numbers correspond with the values in (A). C: A heat map representation of the late window firing rate. D: A heat map of the linear additions of the firing rates to unimodal visual stimuli (left most column) and auditory stimuli (bottom row). E: Difference of the firing rates in response to multisensory stimuli (C) from linear addition of the unimodal firing rates (D). This neuron exhibits superlinear addition when the auditory stimulus is weak and sublinear addition when the auditory stimulus is strong. F-I: Modeled responses for this neuron. The model captures complex non-linear responses to different intensity levels, as well as a shift from superlinear regime to sublinear regime as a function of the auditory intensity. [quantification of the modulation coefficient (showing alpha, for example)].

as spatial alignment of different sources (or surround suppression in other contexts[21,22]); the second property is known as the principle of inverse effectiveness[1]. These two previously described phenomena are now quantified with one modulation function, and our model-based analysis finds them heuristically, without prior knowledge of the system.

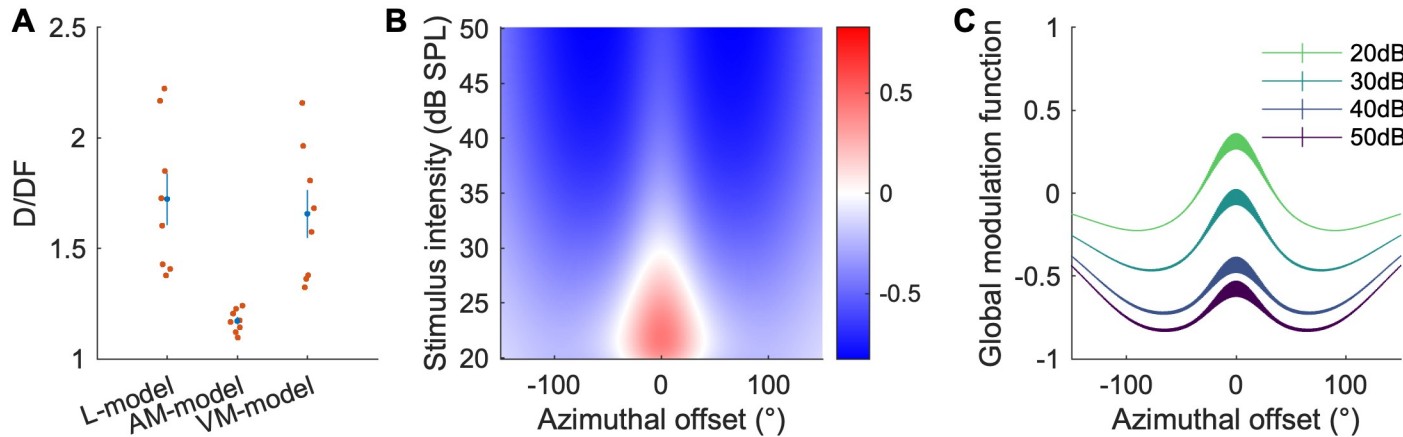

**Fig 6. Global fit statistics of multisensory processing models.** A: The deviance per degree of freedom (D/DF) values of each model. Red dots are individual experiments. Blue is the mean and SEM. B: Average global modulation function $\delta(\phi_A, I_A)$ from all the experiments. Superlinear addition could only occur in the area indicated by red, which is restricted to the small azimuthal offset range and the low auditory stimulus intensity. C: Same data as B for specific auditory intensities as a function of the azimuthal offset.

### The AM-model improves the identification of neurons with multisensory processing

The utility of modeling is that it allows us to compress data from a high-dimensional stimulus space into a low-dimensional model parameter space. This overcomes the difficulty of analyzing and testing the significance of the neurons' response properties created by the large number (168) of visual and auditory stimulus patterns to identify those that exhibit multisensory interactions (see Materials and methods). For example, we used four variables—the locations and intensities of the visual and auditory stimuli—to make our stimuli (Fig 4), thus creating a four dimensional space and 168 different patterns of stimuli. The resulting data are not only difficult to visualize but also suffer from the statistical issues of multiple comparisons.

By using the AM-model to identify neurons that exhibit multisensory interactions we find that 47% (190 / 403) of the modeled neurons exhibit a significant modulation weight ($\alpha > 1$ with p < 0.01) including 17% (70 / 403) of the neurons with superlinear addition ($\alpha^\delta > 1$ with p < 0.01; example: Figs 4, 7C and 7D) and 13% (53 / 403) neurons with sublinear addition ($\alpha^\delta$

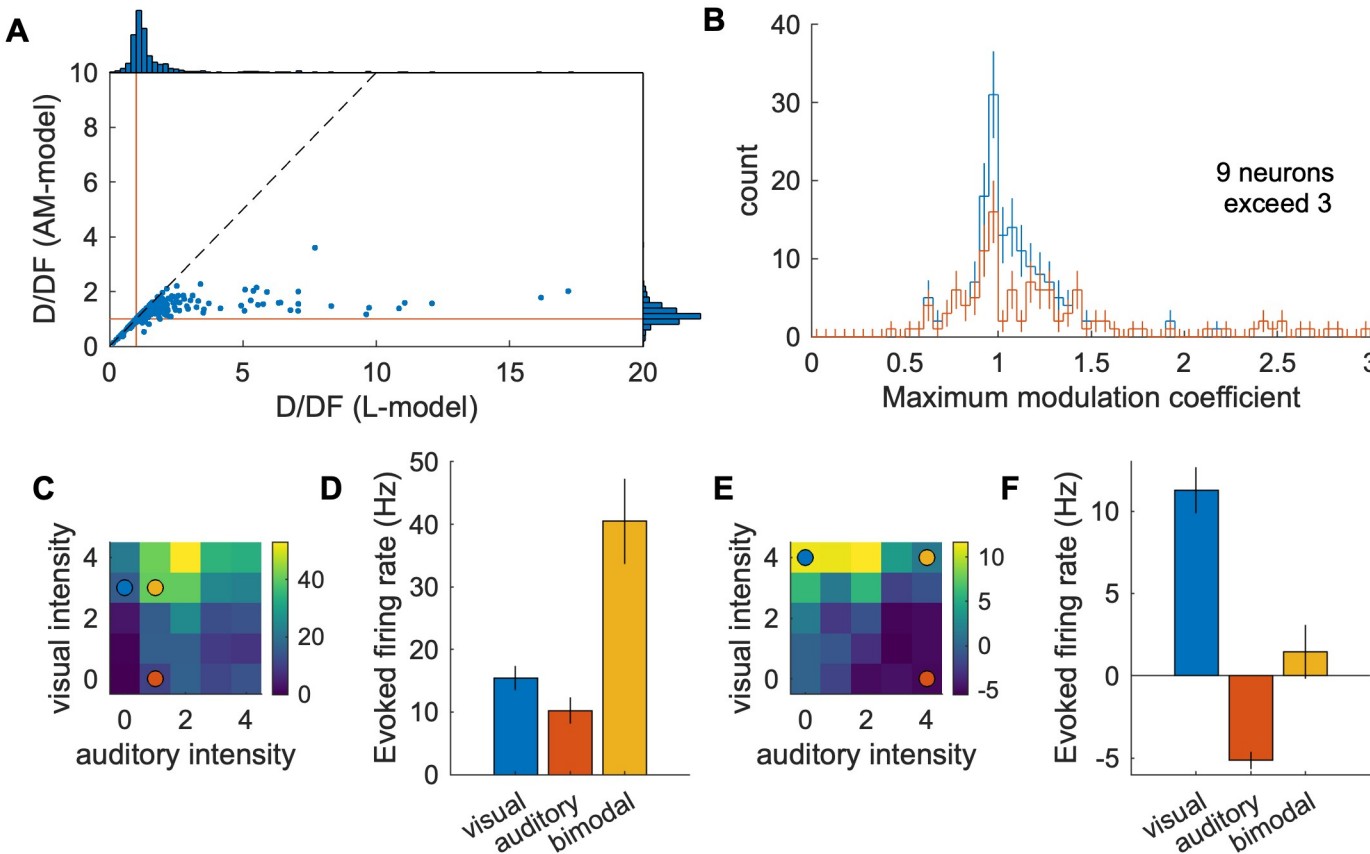

**Fig 7. Visual/Auditory interactions in the mouse SC.** A: Deviance per degree of freedom (D/DF) of individual neurons. Red lines indicate D/DF = 1, good-fit lines. Many of the high D/DF neurons (bad-fit neurons) with the L-model have much smaller D/DF with the AM-model. There are few neurons that have high D/DF with the AM-model compared to the L-model—for example, there is only one neuron with D/DF > 3 in the AM-model, while there are 33 in the L-model—indicating that the AM-model is a good description of the data. B: Histogram of the maximum modulation coefficient ($max(\alpha^\delta)$) of 190 neurons with significant modulation weight ($\alpha > 1$ with p < 0.01). The red line is a histogram of neurons with the maximum modulation coefficient significantly (p < 0.01) different from 1. The modulation coefficient larger (less) than 1 indicates that the neuron exhibited superlinear (sublinear) addition of the visual and auditory responses. C, D: Responses of an example neuron that exhibits superlinear multisensory integration viewed as a heat map (C) and with the firing rate of selected combinations of stimuli (colored dots) graphed in (D). A yellow circle in (C) indicates a combination of stimulus intensities with which significant superlinear enhancement is observed. Blue and red circles indicate corresponding unimodal stimuli. The details of the response of this neuron is elaborated in Fig 5. E, F: Responses of an example neuron that exhibits multisensory suppression viewed as a heat map (E) and with the firing rate of selected combinations of stimuli (colored dots) graphed in (F). Colored circles indicate stimulus combinations with which multisensory suppression is observed.

< 1 with p < 0.01; example: Fig 7E and 7F) when the modulation coefficient ($\alpha^\delta$) is maximized. The distribution of maximum $\alpha^\delta$ for significantly modulated neurons is shown in Fig 7B.

Finding neurons that exhibit superlinear multisensory enhancement is difficult without a well-fit model; naively trying to find superlinear multisensory enhancement would require a significance test for all 144 combinations of the visuoauditory stimuli, and thus would suffer from a multiple comparison correction. For example, we found that only 4 neurons passed this significance test (p < 0.01 / 144). These results demonstrate the difficulty in the detection of multisensory enhancement using generic stimuli and how the model-based analysis overcomes this difficulty by connecting the population-level description of the modulation to an individual neuron's sensory integration.

## Discussion

In this study, we present the first evidence of nonlinear multisensory integration (both enhancement and suppression) in the mouse SC, and demonstrate that the nonlinearity of the response depends on early auditory processing that modulates later visual/auditory responses. To achieve this, we used large-scale recording and a model-based analysis to quantify and visualize two principles of sensory integration that have been previously proposed based on studies of individual neurons: the spatial alignment of RFs and the principle of inverse effectiveness[1].

### Topographic organization of visual, auditory, and bimodal neurons in the SC

The organization of sensory inputs into topographic maps and the subsequent integration of these maps in associative areas of the brain is a fundamental feature of neural processing. We confirm results from previous studies which show that the visual map of azimuthal space in the sSC is aligned with the auditory map of space in the dSC[17–19] and extend this finding to show that most of the visuoauditory bimodal neurons in the mouse SC have overlapping visual and auditory spatial RFs that themselves form a topographic map of space. This supports the model that topographic organization is used to align spatial visual and spatial auditory information perhaps to facilitate the integration of these two modalities.

### Multisensory enhancement and inhibition in the mouse SC

We find that the mouse SC contains a significant percentage of stimulus-responsive neurons that exhibit superlinear multisensory enhancement (~17%) and sublinear addition (~13%; Fig 7B). A similar distribution of nonlinearity has been reported in the barn owl tectum[13]. In some previous studies the responses of single neurons are characterized by tailored stimuli (meaning that the time delay between the visual and auditory stimuli were adjusted for each neuron) to increase the yield of multisensory integration. Instead, we performed large scale recordings that presented the same stimuli to all neurons as happens in the real world. This approach requires many different patterns of stimuli and to prevent the data from being sacrificed by a multiple comparison correction, we used a model-based approach. As large-scale recording methods are becoming more common[23–25], we believe that similar model based approaches should become the norm instead of using an ad-hoc index (such as multisensory index [26] or orientation selectivity index [16,27]) for each application.

### Early auditory processing modulates the visuoauditory response

We present multiple results that suggest that early auditory processing induces visuoauditory integration in the SC: (1) only auditory responses arise in the early ($< 20$ ms) time window; (2) in the late ($> 20$ ms) time window, auditory responses are non-monotonic, while visual responses are monotonic; and (3) nonlinearity of the multisensory integration depends on the properties of auditory stimuli, but not visual stimuli. This result is consistent with a study in goldfish Mauthner cells in which the level of multisensory integration depends on the auditory stimulus more strongly than the visual stimulus [28]. The Mauthner cells receive inputs from the optic tectum and are associated with the escape reflex, which the SC is also involved in. However, we did not identify the source of this modulation in this study. Potential sources include local circuitry within the SC[21,29], a pathway with larger latency such as a cortical pathway[30], and/or reciprocal connections with other brain areas such as the parabigeminal nucleus[31–33].

### Conclusion

The auditory and visual RFs of the neurons in the mouse SC are overlapping and organized as a spatial topographic map and visuoauditory neurons exhibit nonlinear MSI. A model that extends a simple sigmoidal response with an audition-dependent modulation function can explain 76% of the outlier deviance of the simple sigmoidal model. The resulting modulation function is consistent with two important principles of multisensory integration: spatial matching and inverse effectiveness. These results open the door to further studies designed to determine the circuitry underlying multisensory integration, and the mechanisms used for its development.

## Materials and methods

### Ethics statement

The care of animals in this study was carried out in accordance with the University of California, Santa Cruz Institutional Animal Care and Use Committee protocol number Feld2011.

### Animal preparation for electrophysiology

Animal preparation, head plate attachment and craniotomy were performed as previously described[17]. We used 2–9 month-old CBA/CaJ (The Jackson Laboratory, 000654) mice of each sex. One to four days before the recording, a custom-made titanium head plate was attached to the mouse's skull. On the day of the recording, the mouse was anesthetized with isoflurane (3% induction, 1.5–2% maintenance; in 100% oxygen) and a craniotomy was made (~1.5–2 mm diameter) in the left hemisphere above the SC (0.6 mm lateral from the midline, on the lambdoid suture). A 256-channel silicon probe (provided by Prof. Masmanidis [25,34]) was inserted through the cortex into the SC with its four shanks aligned along the A–P axis (Fig 1A). The probe was lowered until robust multi-unit visual responses from the superficial SC reached the top of the active region of the most posterior shank. The probe was then retracted by 120–200 μm from its location to reduce drifting during the recording. The mice were euthanized after the recording session.

During the recording, the mouse was allowed to run freely on a cylindrical treadmill made of polystyrene foam. The movement of the cylinder was recorded by a shaft encoder (US Digital, H5-100-NE-S).

## Stimulation

**Visual stimulation.** To measure visual receptive fields (RFs), a 10˚-diameter flashing circular spot on a $16 \times 7$ grid with 10˚ spacing[16] was presented from a computer monitor (Samsung S29E790C; 67.3 cm x 23.4 cm active display size; refresh rate: 60 Hz; gamma corrected) placed 25 cm away from the animal. The stimuli were presented using a custom Matlab program that uses the Psychophysics Toolbox extensions[35–38]. The 500-ms flashes were either ON (white) or OFF (black) on a gray background at mean luminance (32 cd/m$^2$) and a 500-ms gray screen was inserted after each stimulus presentation. The stimulus was presented in a random order of the contrast and location. Every pattern was repeated 12 times. To estimate the RF parameters, a 2-dimensional Gaussian was fit to the collected neural responses (unbinned maximum-likelihood estimation[16]).

To collect the local field potentials (LFPs) used to estimate the surface location of the SC [16], a contrast-alternating checkerboard stimulus (0.04 cpd square wave alternating at 0.5 Hz) was presented.

**Auditory stimulation.** Virtual auditory space stimulation for mice was done in an anechoic chamber as previously described[17]. Briefly, sound is produced through earphones that are placed near the mouse's pinnae. The incident direction of the sound was controlled using filters derived from the head-related transfer functions of the mouse.

To measure the auditory receptive fields of the neurons, a full-field white-noise burst was presented from each virtual location in a grid space of 5 elevations (0˚ to 80˚ with 20˚ steps) and 17 azimuths (-144˚ to 144˚ with 18˚ steps), totaling 85 points in a two-dimensional directional space. A maximum-likelihood fit of the Kent distribution was used to determine the spatial properties of the auditory receptive fields[17]. Because the speakers for auditory stimulation blocked a part of the visual field where the azimuth is larger than 90˚, we excluded neurons with RF azimuth larger than 90˚ (either visual or auditory) from the analysis.

**Multisensory stimulation.** To create a multisensory stimulation system the above visual and auditory stimulation conditions were combined with the visual stimulation computer used as a master computer for synchronization. To synchronize the timing of the stimuli, the visual stimulation computer sends a transistor-to-transistor logic (TTL) pulse to the auditory stimulation computer, and the auditory stimulation computer sends a TTL pulse to the data acquisition system (RHD2000, Intan Technologies). Because the entire system did not fit in our anechoic chamber, this experiment was done without an anechoic chamber after verifying that the auditory RFs of the SC neurons are not significantly altered by removing the anechoic chamber.

Visuoauditory spot stimulation was used to measure the properties of multisensory processing. The duration of the visual and auditory stimuli were 100 ms. The auditory stimulus is delivered ~13 ms later than the visual stimulus, mimicking the effect of the time delay due to the slow propagation speed of sound in the air (343 m/s; compared to the speed of light). This time delay is consistent with a simultaneous visual and auditory event occurring at 4–5 meters away from the animal. Temporal responses of neurons are calculated relative to the onset of the auditory stimulation. The intensity of the black spots were (0, -0.29, -0.50, -0.65, -0.75) in two of the experiments, and (0, -0.1, -0.2, -0.4, -0.8) in the other 6 experiments, measured as a fractional change from the gray background (for example, -0.1 is 10% reduction of the brightness level, or 28.8 cd/m$^2$ compared to the 32 cd/m$^2$ background). The intensity of the auditory stimuli (white noise bursts) were 20, 30, 40, and 50 dB SPL (within 5–80 kHz), or no sound. The stimuli were presented at three locations, 30 degrees apart, that match the visual RF positions of 3 of the silicon probe shanks. We presented black spots because black spots excite more SC neurons than white spots[39]. The total number of the patterns presented to the

animal are 5 (visual intensities) × 5 (auditory intensities) × 3 (visual locations) × 3 (auditory locations) = 225 patterns. These patterns were presented in a random order in a 2-s period and repeated 30 times. When either no visual or auditory stimulus is given, this stimulus becomes unimodal, and is used for intensity tuning analysis, as described in the section on "Visual and early auditory responses are monotonic; late auditory responses can be non-monotonic".

## Data analysis

**Blind analysis.** We used blind analysis, an effective method for reducing false-positive reporting of results[40,41]. Of the 8 recording data sets, 4 were used for exploratory purposes and the other 4 were blinded. Once the features and parameters to be analyzed were decided the same analysis was done on the blinded dataset to confirm if the results agreed with those of the exploratory dataset. All of the results reported in the present study passed a significance test both in the exploratory dataset and the blinded dataset. The combined dataset was used to derive the p-values. Of the individual datasets shown in S3 Fig, data A-D are exploratory datasets and data E-H are blinded datasets.

**Spike-sorting.** Spikes were sorted using custom-designed software[16,17,42]. Briefly, raw analog signals were high-pass filtered (cutoff ~313 Hz, wavelet filter[43]) and thresholded for spike detection. Detected spikes were clustered employing principal component analysis to its waveforms on the seed electrode and its surrounding electrodes. A mixture-of-Gaussians model was used for cluster identification. Clusters with many spikes in the refractory period and duplicated clusters for the same neuron (determined by isolation distance, L-ratio, cross-correlation analysis and similarity of the waveform shapes) were excluded from the analysis. Neurons with axonal waveforms were identified by principal component analysis and excluded from the analysis[17].

**Estimation of the relative position of the neurons across datasets.** To make a map of visual/auditory space across datasets from multiple mice, the relative physical locations of the SC neurons were estimated using the visual RFs. The visual RFs on multiple shanks were used to estimate the A–P position where the visual RF azimuth was 0˚, and this point was defined as the zero point of the A–P SC position. The height of the SC surface is estimated by LFPs in response to a checkerboard visual stimulus[16]. Estimating that the insertion angle of the probe into the SC was 25˚[17], the A–P positions and the depths of the neurons across multiple datasets were calculated and superimposed. Only neurons with a positive A–P position were analyzed in order not to include neurons outside the SC.

**Model of multisensory responses.** Our model comprises commonly used sigmoidal intensity tuning curves and an auditory-dependent modulation term derived from the analysis of the multisensory responses (AM-model, see Results). We formulated our multisensory model as follows.

For each neuron, the response $R$ to a visuoauditory stimulus is determined by the following equations.

$$R(\phi_V, \phi_A, I_V, I_A) = \alpha^{\delta(\phi_A, I_A)}\left(F_V(\phi_V)S(I_V; I_{V50}, n_V) + F_A(\phi_A)S(I_A; I_{A50}, n_A) + b\right)$$

where,

$$S(\phi_A, I_A) = A_{Exc}M_{Exc}(\phi_A, I_A) - A_{Inh}M_{Inh}(\phi_A, I_A),$$

$$S(I; I_{50}, n) = \frac{I^n}{I_{50}{}^n + I^n},$$

and,

$$M_T(\phi_A, I_A) = G(\phi_{eff} - \phi_A, \sigma_T) S(I_A; I_{T50}, n_T$$

$R(\phi_V, \phi_A, I_V, I_A)$ is the resulting firing rate of a neuron when a visual stimulus is presented at the azimuth $\phi_V$ with the intensity $I_V$ and an auditory stimulus is presented at the azimuth $\phi_A$ with the intensity $I_A$; $G(\mu, \sigma)$ is a Gaussian with a unit height, mean $\mu$ and standard deviation $\sigma$; $T$ is a sign of the modulation (*Exc* (excitatory) or *Inh* (inhibitory)), $\phi_{eff}$ is the neuron's effective RF azimuth (we used this instead of measured RF azimuths because some neurons do not have a measured RF azimuth); and $\sigma$ is the width of the modulation effect along the azimuthal axis. The baseline (spontaneous) firing rate $b$ is a fixed value that is measured from all the intervals of the stimuli and is not a model parameter. The descriptions and the ranges of the model parameters are summarized in Table 2. Parameters labelled as "Individual" apply independently to each neuron; Global parameters apply to all the neurons in a given dataset.

To test the model without modulation (L-model), we set $\alpha = 1$. By setting this value, the number of the global parameters go down from 8 to 0, and the number of the individual parameters go down from 12 to 10 ($\alpha$ and $\phi_{eff}$ become irrelevant). To test vision-dependent modulation (instead of auditory-dependent modulation; VM-model), we changed the power of the modulation term to the following.

$$\delta(\phi_V, I_V) = A_{Exc} M_{Exc}(\phi_V, I_V) - A_{Inh} M_{Inh}(\phi_V, I_V)$$

Although the total number of stimulus patterns are 225 as noted above, some of the patterns are redundant. Namely, when either visual or auditory stimulus intensity is zero, its location becomes an irrelevant variable. Thus, the number of non-redundant stimulus patterns are 144 (4 x 4 x 3 x 3; multisensory stimuli) + 12 (4 x 3, no visual stimuli) + 12 (4 x 3, no auditory stimuli) = 168. Therefore, the number of degrees of freedom is $168N - (12N + 8) = 156N - 8$, where N is the number of neurons included in the analysis. As the modulation function is not relevant for the L-model, its number of degrees of freedom is $168N - 10N = 158N$.

**Fitting procedure of the multisensory model.** A maximum-likelihood method was used to fit our model to the data. The likelihood function—deviance (*D*: twice the negative log-likelihood (NLL) ratio)—for one neuron is defined based on quasi-Poisson statistics[17,44,45] as follows.

$$D = \sum_{\hat{V}} \frac{2R_{obs}(\hat{V})}{\theta(\hat{V})} \left( \frac{R_{model}(\hat{V})}{R_{obs}(\hat{V})} - 1 - log \frac{R_{model}(\hat{V})}{R_{obs}(\hat{V})} \right)$$

where $\hat{V}$ represents the four stimulus variables $\{\phi_V, \phi_A, I_V, I_A\}$, $\theta$ is a dispersion factor defined as $\theta(\hat{V}) = \frac{\sigma_{obs}^2(\hat{V})}{R_{obs}(\hat{V})}$, $R_{obs}$ is the observed mean firing rate over trials, $\sigma_{obs}$ is the SEM of the observed firing rate, and $R_{model}$ is the estimated firing rate by the model described above. The sum is over all 168 non-redundant combinations of the stimulus patterns (see previous section). Deviance for the entire dataset is a sum of the individual deviances of all the modeled neurons.

Because of the global parameters that affect all the neurons, the fitting must be done simultaneously for all the neurons. In order to achieve the best accuracy of the modulation function, we included all the neurons that had either visual or auditory responses in the late time window. Since there are tens of responsive neurons per dataset, hundreds of parameters must be simultaneously fit to thousands of data points. We minimized the effort of this challenging fitting procedure by breaking down the function fit into smaller problems with a smaller number

**Table 2. Summary of the model parameters.**

| Use of the parameters | Number of parameters | Individual or global | Range (Phase 1) | Range (Phase 2) |
|---|---|---|---|---|
| $F_V(\phi_V)$: Max visually evoked firing rates | 3 (one per stim location) | Individual | (0–100) Hz | (-100-200) Hz |
| $I_{V50}$, $n_V$: Visual intensity response sigmoid parameters | 2 | Individual | $c_{V50}$: 0.01–1<br>$n_V$: 0–5 | $c_{V50}$: 0.01–3<br>n: 0–30 |
| $F_A(\phi_A)$: Max auditory evoked firing rates | 3 (one per stim location) | Individual | (0–100) Hz | (-100-200) Hz |
| $I_{A50}$, $n_A$: Auditory intensity response sigmoid parameters | 2 | Individual | $c_{A50}$: 0.1–100<br>$n_A$: 0–5 | $c_{A50}$: 0.1–300<br>$n_A$: 0–30 |
| $\alpha$: Modulation weight | 1 | Individual | 1.01–30 | 1–50 |
| $\phi_{eff}$: effective RF azimuth of the neuron | 1 | Individual | A fixed value estimated from the neuron's physical location | (-100-150) ° |
| $\sigma_T$: Width of modulation | 2 (excitatory & inhibitory) | Global | (18–100) ° | (18–100) ° |
| $I_{T50}$, $n_T$: Intensity response sigmoid for modulation | 4 (excitatory & inhibitory) | Global | $c_{T50}$: 0.1–100<br>$n_T$: 0.1–5 | $c_{T50}$: 0.1–200<br>$n_T$: 0.1–5 |
| $A_T$: Relative amplitude of modulation | 2 (excitatory & inhibitory) | Global | Exc: 0–10<br>Inh: 0–10 | Exc: 0–20<br>Inh: 0–20 |
| Total number of parameters | 12 * N + 8 | | | |

of parameters and repeating simple function fits. Namely, when the individual parameters of a neuron are changed, the global parameters are fixed, so the problem is reduced to 12 parameters for the neuron, and when the 8 global parameters are changed, the individual parameters for the neurons are fixed. These fittings were alternately repeated until the overall likelihood value converged (the exact condition is written below).

The fit was performed in two phases because doing the entire fit at once was unstable. For example, if both the neurons' effective RF azimuth ($\phi$), and the modulation shapes are changed simultaneously, too many local minima are created and the fit cannot escape from them. By fixing the azimuths of the RF location in the first phase, we could stabilize the resulting properties of the modulation function (and its likelihood value was consistently better). In the second phase, some of the parameter boundaries were also relaxed (see Table 2).

The detailed fitting procedure is the following.

Phase 1: (narrow parameter boundaries)

1. Decide initial visual and auditory sigmoid parameters for individual neurons using unisensory response data.

2. Randomly set the modulation parameters in their allowable ranges (except for $A_T$ that are set to a small value (0.1) to stabilize fitting).

3. Refit individual neurons using the full data and modulation.

4. Refit the modulation parameters using the updated individual cell parameters.

5. Repeat 3–4 until the change of NLL in one iteration is below the tolerance value (0.1).

6. Once it reaches the tolerance, perform a detailed error analysis (MINOS[46]) to further improve the fit and repeat 3–4 again.

7. Stop if the likelihood gain is less than the tolerant value after the MINOS analysis.

Phase 2: (relaxed parameter boundaries)

1. Perform the same procedure as 3–7 of Phase 1 with the relaxed parameter boundaries.

We repeated this fitting procedure 30 times with random initial modulation parameters, and used the fit that minimized the NLL value. Once the fitting is completed, the error of each parameter was estimated from the Hessian matrix of this likelihood function. To calculate goodness-of-fit of individual neurons, the asymptotic equivalence between $\chi^2$ and two times NLL (Wilks' theorem[47]) was assumed.

## Supporting information

**S1 Fig. Sequence and timing of events (stimuli and responses) in these multisensory experiments.** The 13 ms delay in the arrival of the auditory stimulus at the ears is based on a 4.5 m sound source distance and a sound speed of 343 m/s.
(TIF)

**S2 Fig. Example responses of visuo-auditory bimodal neurons to changes in illumination and sound level.** A: Raster plots of two example neurons to different intensities of monomodal visual and auditory stimuli. (NS indicates no stimulus.) These neurons exhibit non-monotonic auditory responses in the late time scale. The red line indicates 20 ms that separates the early and late time windows of auditory responses. B: Summary firing rates of the two neurons in (A). The stimulus intensities are indicated by 0–4 for brevity; their actual values correspond with those used in (A). These neurons' peak early auditory response is not significantly larger than their max-intensity response (p = 0.25 for neuron 1; p = 0.13 for neuron 2), therefore, these are not considered 'non-monotonic' responses. On the other hand, the peak late-auditory response is significantly larger than the max-intensity response (p = 4 x $10^{-10}$ for neuron 1; p = 0.004 for neuron 2), therefore, these are considered 'non-monotonic' responses.
(TIF)

**S3 Fig. Global modulation functions of each individual dataset.** Datasets A-D were used as the exploratory datasets while E-H were used as the blinded datasets (see Materials and methods). N: Number of neurons with late visual or late auditory responses in each dataset. D/DF: Deviance per degree of freedom.
(TIF)

## Acknowledgments

We thank Sotiris Masmanidis for providing us with the silicon probes; and Jena Yamada for helpful comments on the manuscript.

## Author Contributions

**Conceptualization:** Shinya Ito.

**Data curation:** Shinya Ito.

**Formal analysis:** Shinya Ito, Yufei Si.

**Funding acquisition:** Alan M. Litke, David A. Feldheim.

**Investigation:** Shinya Ito, Yufei Si.

**Methodology:** Shinya Ito.

**Resources:** David A. Feldheim.

**Supervision:** Alan M. Litke, David A. Feldheim.

**Writing – original draft:** Shinya Ito.

**Writing – review & editing:** Yufei Si, Alan M. Litke, David A. Feldheim.

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
