## [Decision Letter · Decision Letter 0]

18 Jul 2021

Dear Dr. Feldheim,

Thank you very much for submitting your manuscript "Nonlinear visuoauditory integration in the mouse superior colliculus" for consideration at PLOS Computational Biology.

As with all papers reviewed by the journal, your manuscript was reviewed by members of the editorial board and by several independent reviewers. In light of the reviews (below this email), we would like to invite the resubmission of a significantly-revised version that takes into account the reviewers' comments. In particular, both reviewers questioned the definition of "monotonic" vs. "non-monotonic", and the reviewer 2 raised a couple of important issues that should be addressed in the revised manuscript.

We cannot make any decision about publication until we have seen the revised manuscript and your response to the reviewers' comments. Your revised manuscript is also likely to be sent to reviewers for further evaluation.

Sincerely,

Tianming Yang

Associate Editor

PLOS Computational Biology

Wolfgang Einhäuser

Deputy Editor

PLOS Computational Biology

Reviewer's Responses to Questions

**Comments to the Authors:**

Reviewer #1: The authors studied the response properties of mouse SC neurons to spatially restricted visual and auditory stimuli using large-scale physiology. Their data reveal some very important and exciting findings, such as that the mouse SC contains topographically organized visual and auditory neurons; the two visual and auditory maps are aligned; and many neurons exhibit nonlinear multisensory integration. These findings have the potential to form the foundation for future studies on these topics and will be highly interesting to researchers in the field. I have only some suggestions to improve presentation.

1. Unless there is a limit to the number of figures, I suggest that some of the figures be split and some of the supplementary figures moved to main figures to help describe and support the findings.

2. For example, Fig. 1 can be separated into at least 2 figures, with the receptive field data more extensively presented. Quantification of Fig. 1g for all cells that responded to both stimuli, in terms of size, overlap, position correspondence, and depth profile.

3. Fig. 2, how is monotonic response defined/quantified? A detailed explanation is needed in the main text.

4. A major finding of the paper is about non-linear integration between auditory and visual responses, but no examples are shown in the main figure. I suggest moving much of supplementary figure 4 to a main figure, before the model. I understand the statistical issues of conventional approaches to quantify non-linear interactions, but maybe you can use the “best response” to auditory and visual combinations, and then see if that is supralinear or sublinear compared to responses to single stimuli? Better yet, compare this conventional method with the modeling approach to illustrate the model’s effectiveness and superior performance.

5. Fig. 3 and the associated model description are a bit too opaque. Maybe use some example cells to illustrate this modeling approach to help readers to understand?

Reviewer #2: In the current study, the authors examined visual, auditory, and combined responses of SC neurons in mice. The main finding is that there are many cases showing nonlinear visuo-auditory integration in addition to linear integration. And the authors show that this nonlinear integration factor mainly comes from the auditory modulation, instead from visual. The results are clear. The paper is well written. Here are a few comments and queries.

1. What is the criterion to group “monotonic” and “non-monotonic”?

2. The terms of “monotonic” and “non-monotonic” are not proper, because both are monotonic, but just that one is positive and the other is negative as a function of stimulus intensity. Plus, from a few example tuning curves shown, they are not really monotonic, especially for the auditory, by reaching a plateau.

3. The authors show that early auditory processing modulates visuoauditory integration, yet through their model, it seems that it can also modulate the visual tuning itself, right? If so, how can visual tuning, especially the monotonic tuning is modulated by auditory early component? Something is puzzling here.

4. In Fig. 3cd, modulation coefficient refers to δ, whereas in Fig.4b, modulation coefficient refers to α^δ. Suggest to use different terms.

5. There appear to be an interaction between the auditory early and late component. However, the authors did not show these data clearly. I would like to see a direction comparison, by a scatter plot, between early and late auditory responses for each neuron. The scatter plot needs to be made for each stimulus intensity, leading to 4-5 plots.

**Have the authors made all data and (if applicable) computational code underlying the findings in their manuscript fully available?**

Reviewer #1: Yes

Reviewer #2: Yes

PLOS authors have the option to publish the peer review history of their article (what does this mean?). If published, this will include your full peer review and any attached files.

Reviewer #1: No

Reviewer #2: No
---

## [Decision Letter · Decision Letter 1]

4 Oct 2021

Dear Dr. Feldheim,

We are pleased to inform you that your manuscript 'Nonlinear visuoauditory integration in the mouse superior colliculus' has been provisionally accepted for publication in PLOS Computational Biology.

Best regards,

Tianming Yang

Associate Editor

PLOS Computational Biology

Wolfgang Einhäuser

Deputy Editor

PLOS Computational Biology

Reviewer's Responses to Questions

**Comments to the Authors:**

Reviewer #1: The authors have adequately addressed my comments.

Reviewer #2: The authors have addressed my points. I do not have further comments.

**Have the authors made all data and (if applicable) computational code underlying the findings in their manuscript fully available?**

Reviewer #1: None

Reviewer #2: Yes

PLOS authors have the option to publish the peer review history of their article (what does this mean?). If published, this will include your full peer review and any attached files.

Reviewer #1: No

Reviewer #2: No

---

## [Editor Report · Acceptance letter]

15 Oct 2021

PCOMPBIOL-D-21-01124R1 

Nonlinear visuoauditory integration in the mouse superior colliculus

Dear Dr Feldheim,

I am pleased to inform you that your manuscript has been formally accepted for publication in PLOS Computational Biology. Your manuscript is now with our production department and you will be notified of the publication date in due course.

With kind regards,

Olena Szabo
